# Association between the Right Ventricular Longitudinal Shortening Fraction and Mortality in Acute Respiratory Distress Syndrome Related to COVID-19 Infection: A Prospective Study

**DOI:** 10.3390/jcm11092625

**Published:** 2022-05-06

**Authors:** Christophe Beyls, Camille Daumin, Alexis Hermida, Thomas Booz, Tristan Ghesquieres, Maxime Crombet, Nicolas Martin, Pierre Huette, Vincent Jounieaux, Hervé Dupont, Osama Abou-Arab, Yazine Mahjoub

**Affiliations:** 1Department of Anesthesiology and Critical Care Medicine, Amiens University Hospital, F-80054 Amiens, France; daumin.camille@chu-amiens.fr (C.D.); booz.thomas@chu-amiens.fr (T.B.); ghesquieres.tristan@cgu-amiens.fr (T.G.); crombet.maxime@chu-amiens.fr (M.C.); huette.pierre@chu-amiens.fr (P.H.); dupont.herve@chu-amiens.fr (H.D.); abouarab.osama@chu-amiens.fr (O.A.-A.); mahjoub.yazine@chu-amiens.fr (Y.M.); 2UR UPJV 7518 SSPC (Simplification of Care of Complex Surgical Patients) Research Unit, University of Picardie Jules Verne, F-80000 Amiens, France; 3Department of Cardiology, Amiens University Hospital, F-80054 Amiens, France; hermida.alexis@chu-amiens.fr (A.H.); martin.nicolas@chu-amiens.fr (N.M.); 4Respiratory Department, Amiens University Hospital, F-80054 Amiens, France; jounieaux.vincent@chu-amiens.fr

**Keywords:** RV-LSF, right ventricle, speckle-tracking, ARDS, COVID-19

## Abstract

Introduction: Right ventricular systolic dysfunction (RVsD) increases acute respiratory distress syndrome mortality in COVID-19 infection (CARDS). The RV longitudinal shortening fraction (RV-LSF) is an angle-independent and automatically calculated speckle-tracking parameter. We explored the association between RV-LSF and 30-day mortality in CARDS patients. Methods: Moderate-to-severe CARDS patients hospitalized at Amiens University Hospital with transesophageal echocardiography performed within 48 h of intensive care unit admission were included. RVsD was defined by an RV-LSF of <20%. The patients were divided into two groups according to the presence of RVsD. Using multivariate Cox regression, clinical and echocardiographic risk factors predicting 30-day mortality were evaluated. Results: Between 28 February 2020 and 1 December 2021, 86 patients were included. A total of 43% (*n* = 37/86) of the patients showed RVsD and 22% (*n* = 19/86) of the patients died. RV-LSF was observed in 26 (23.1–29.7)% of the no-RVsD function group and 16.5 (13.7–19.4)% (*p* < 0.001) of the RVsD group. Cardiogenic shock (*n* = 7/37 vs. 2/49, *p* = 0.03) and acute cor pulmonale (*n* = 18/37 vs. 10/49, *p* = 0.009) were more frequent in the RVsD group. The 30-day mortality was higher in the RVsD group (15/37 vs. 4/49, *p* = 0.001). In a multivariable Cox model, RV-LSF was an independent mortality factor (HR 4.45, 95%CI (1.43–13.8), *p* = 0.01). Conclusion: in a cohort of moderate-to-severe CARDS patients under mechanical ventilation, RVsD defined by the RV-LSF was associated with higher 30-day mortalities.

## 1. Introduction

Right ventricular (RV) systolic dysfunction (RVsD) is a common echocardiographic feature in COVID-19 infection and is associated with increased mortality [1]. Using echocardiography, RVsD can be measured by conventional or advanced bi-dimensional RV speckle-tracking parameters [2]. In acute respiratory distress syndrome (ARDS) related to COVID-19 infection (CARDS), the evaluation of RV systolic function is crucial for ventilation settings adaptation, hemodynamic status evaluation, and fluid balance management [3]. In ARDS clinical situations with patients under mechanical ventilation, transesophageal echocardiography (TEE) is more accurate than transthoracic echocardiography for RV systolic function assessment [4]. In TEE, RV systolic function is mainly assessed by the RV fractional area change (RV-FAC), even though the use of RV speckle-tracking parameters, such as strain parameters, is booming [5]. However, the prognosis of RV systolic dysfunction assessed by strain parameters is subject to ongoing debate [6,7]. 

RV longitudinal shortening fraction (RV-LSF) is a recent semi-automatic bi-dimensional speckle-tracking echocardiographic (2D-STE) parameter based on tricuspid annular displacement, which allows extensive assessment of RV global systolic function [8]. It is an angle independent, reproducible and accurate parameter. Compared to RV strain parameters, RV-LSF is less dependent on loading conditions [9] and image quality [10]. A previous study showed that the RV-LSF cut-off value of 20%, measured in TEE, was accurate for RV systolic dysfunction detection in CARDS [10]. To date, the prognostic value of RV systolic dysfunction assessed by RV-LSF for CARDS patients has not been evaluated. We aimed to explore the association between RV systolic dysfunction, evaluated by RV-LSF, and 30-day mortality in a cohort of moderate-to-severe CARDS patients. We hypothesized that 30-day mortality was higher in the RVsD group.

## 2. Materials and Methods

### 2.1. Population

Adult patients (>18 years of age) with documented COVID-19 infection admitted to our intensive care unit (ICU) for moderate-to-severe CARDS under mechanical ventilation were prospectively included in the study. The exclusion criteria were patients with permanent atrial fibrillation, permanent atrial and ventricular pacing, contra-indications to TEE (esophageal disease or major uncontrolled bleeding), women’s pregnancy, patients under extracorporeal membrane oxygenation (ECMO), and those with poor image quality for 2D-STE parameters analysis. The patients were included on the day when TEE was performed. 

### 2.2. Ethics

This is an ancillary study of a prospective cohort study of patients with COVID-19 infection hospitalized in the ICU at Amiens University Hospital (NCT04354558). The study cohort comprised 29 patients who had been reported in a previous study [10]. 

### 2.3. Data

Data from electronic data and medical reports were collected prospectively. SARS-CoV-2 infection was confirmed by a positive RT-PCR on a nasopharyngeal swab or bronchoalveolar lavage upon admission to our critical care unit. The ARDS grade was defined according to the Berlin definition [11]. The severity of illness upon ICU admission was evaluated with the simplified acute physiology score II and the sequential organ failure assessment (SOFA) score [12]. Vasopressor use was assessed by the SOFA cardiovascular (SOFA cv) score [12] and the vasoactive-inotropic score. Acute kidney injury was defined via the KDIGO classification. In-hospital and thirty-day all-cause mortality were obtained through the Amiens Hospital record database or medical follow-up. 

### 2.4. TEE Measurement 

Trained operators performed TEE in a supine position within 48 h of ICU admission. During TEE examination, all patients were sedated and paralyzed in accordance with ARDS management guidelines [13]. The TEE echocardiography protocol was used, following the American Society of Echocardiography guidelines [5]. Echocardiographic images were obtained by a high-quality, commercially available ultrasound system (CX 50, Philips Healthcare, Andover, MA, USA). All operators had a level III competence of general adult TEE [14]. 

*RV-FAC measurement*: In the four-chamber view at the mid-esophageal level (ME 4CH), RV-FAC was calculated by subtracting the end-systolic area from the end-diastolic area and dividing this value by the end-diastolic area. *Diagnosis of acute cor pulmonale (ACP)*: In a ME 4CH view, the RV end-diastolic area to the left ventricular end-diastolic area ratio was measured, and the septal motion was carefully observed. ACP was defined as the RV end-diastolic area to left ventricular end-diastolic area ratio of >0.6 associated with septal dyskinesia [15]. 

*RV-LSF analysis:* RV-LSF analysis was described in a previous report [10]. For RV-LSF analysis, three points were used to initialize the first diastolic frame in a ME 4CH view (Figure 1A). These points were placed (1) on the tricuspid annulus at the insertion of the anterior tricuspid valve leaflet (RV free wall), (2) on the tricuspid annulus at the insertion of the septal leaflet, and (3) on the RV apex. The software (Automated Cardiac Motion Quantification, QLAB version 9.0, Philips Medical Systems, Andover, MA, USA) automatically tracked and calculated the following three parameters: (1) the displacement between the RV free wall and the RV apex (TAD_lat_), (2) the displacement between the interventricular septum and the RV apex (TAD_sep_) and (3) the RV-LSF. RV-LSF was calculated as the maximum end-systolic displacement (LES) of the mid-annular point from the measured annular motion and is expressed in percent of the end-diastolic RV longitudinal dimension (LED), demonstrated in the following calculation: 100 × (LED—LES)/LED). The mid-annular point was automatically selected by the software (Appendix A). RV systolic dysfunction was defined by an RV-LSF of <20% [10,16].

*RV 2D-strain analysis:* RV strain parameters were obtained using a dedicated software (Automated Cardiac Motion Quantification, QLAB version 13.0 Philips Medical Systems, Andover, MA, USA). RV-strain parameters were obtained by TEE in the ME 4CH view. After defining three points, the region of interest (ROI) was generated automatically and adjusted manually in case of poor quality. RV-FWLS and RV-GLS were calculated automatically by the software (Figure 1B). The longitudinal strain was defined as the percentage of myocardial shortening relative to the original length and presented as a negative value, with a more negative strain value reflecting better shortening [17]. However, for better understanding, the strain parameters were expressed as an absolute value. The 2D-STE parameters were analyzed in a single frame, and the reported values were the average of three measurements. All 2D-STE measurements were performed by an experienced cardiologist blind to the clinical data. 

### 2.5. Statistical Analysis

Data are expressed as mean ± standard deviation (SD), median (interquartile range), or numbers (percentage), as appropriate. The patients were divided in the following two groups: RVsD and non-RVsD group. The variables were compared between groups using Mann–Whitney or Chi-square tests. Univariate and multivariate logistic regression evaluated the independent factors associated with RVsD. Univariate and multivariate COX models were performed to evaluate the independent factors associated with RVsD. All the factors with a *p* value of <0.05 in the univariate analysis were included in the Cox model. We evaluated the prognostic impact of RVsD according to the echocardiographic definition. RVsD was defined as either an RV-FAC of <35%, RV-FWLS of <−20% or RV-LSF of <20% [2,10,18]. The Kaplan–Meier method was used to plot the survival curves between the two groups, which were compared with the logrank test. A statistical test was significant when the *p*-value was under 0.05. All the *p* values are the results of two-tailed tests. The statistical analyses were performed using SPSS software, version 24 (IBM Corp, Armonk, NY, USA).

## 3. Results

Between 28 February 2020 and 1 December 2021, 230 consecutive patients were admitted to our ICU for moderate-to-severe CARDS. Among the 114 patients who underwent the inclusion criteria, 28 patients (24%) were finally non-included and were as follows: twenty-three patients were under extracorporeal membrane oxygenation, one patient was pregnant, one patient had permanent ventricular pacing, and two patients had poor TEE image quality. The study population was divided into two groups according to the presence of RVsD within 48 h of ICU admission. A total of 86 patients were included in the study, with 37 patients in the RVsD group and 49 patients in the no-RVsD group (see Figure 2, flow chart).

Demographic, biological and computed tomography data of the two groups were summarized in Table 1. There were no differences in age, SAPS II score, and medical history between the two groups. In the RVsD group, four patients had a pulmonary embolism (*n* = 4/37 vs. 1/49, *p* = 0.16). There was no difference in ventilator settings between the two groups during TEE (Table 2). The patients had a higher dose of vasopressor administration in the RVsD groups with a higher SOFA_CV_ score. (4 (0–4) vs. 0 (0–4) *p* = 0.02).

For TEE parameters (Table 2), the patients in the RVsD groups had a more dilated RV (RV EDA = 22.4 (18.7–26.7) cm^2^ vs. 18.5 (14.9–22.0) cm^2^, *p* = 0.006), with a more impaired RV-FAC (41.2% (32.0–46.9) % vs. 48.7 (41.1–54.7) %, *p* = 0.005) and left ventricular ejection fraction (LVEF) (56.1 (42.9–67.9) % vs. 65.6 (57.3–72.0) %, *p* = 0.03). ACP was diagnosed in 18 patients (49%) in the RVsD group. Concerning cardiac biomarkers, there was no difference between the ACP and non-ACP group, which were as follows: 19 (11–52) ng·mL^−1^ vs. 39 (16–105) ng·mL^−1^ (*p* = 0.21) for troponine and 41 (21–83) pg·mL^−1^ vs. 59 (25–115) pg·mL^−1^ (*p* = 0.57) for BNP. 2D strain parameters were more impaired in the RVsD group, especially the RV-FWLS (20.2 (16.4–25.6) % vs. 25 (20.5–29.8) %, *p* = 0.002). As it had been defined, RV-LSF was lower in the RVsD group (16.5 (13.7–19.4) % vs. 26.0 (23.0–29.4), *p* = 0.001). 

During the follow-up (Table 3), the patients in the RVsD group had more pulmonary embolisms (*n* = 8/37 vs. *n* = 3/49, *p* = 0.04) and cardiogenic shocks (*n* = 7/37 vs. 2/49, *p* = 0.03). The 30-day mortality rate was significantly higher in the RVsD group than in the no-RVsD group (*n* = 15/37 vs. 4/49, *p* = 0.0001). A post-hoc analysis that considers a type I error (alpha) of 0.05 and is based on mortality incidence found a post-hoc power (beta) of 94.6%.

### Mortality Risk Factors

In the univariate analysis, we tested the different RVsD definitions, and only the RVsD defined by RV-LSF was associated with 30-day mortality (HR = 5.51 CI95% (1.82–16.7), *p* = 0.002). Depending on RVsD definition, the range of patients in the RVsD group varied from 21% to 43% (Appendix A). The multivariable Cox model retained the SAPS II score, ACP, and RV-LSF of <20% as associated with an increased hazard ratio of death (Table 4). 

The Kaplan–Meier survival analysis demonstrated a significant difference between the two groups, according to the presence of RVsD defined by an RV-LSF of <20%. In particular, survival was lower in the RV-LSF of <20% group (logrank test *p*-value = 0.001, Figure 3). 

## 4. Discussion

The results of our study that compare clinical characteristics, outcomes, and 30-day mortality risk factors of moderate-to-severe CARDS patients with RVsD can be summarized as follows: (1) 37 patients (43%) had RVsD, (2) RVsD was not associated with myocardial biomarkers, (3) patients with RVsD had LVEF impairment, (4) RVsD defined by an RV-LSF of <20% was an independent risk factor of 30-day mortality and (5) RV-FAC and RV-FWLS were not associated with 30-day mortality. 

### 4.1. RVsD in CARDS

RVsD is a common finding in COVID-19 disease. In our study, the range of RVsD prevalence varied from 18% to 43%, depending on the RVsD definition used. These results are in accordance with those of Chotalia et al., who demonstrated that 53% of CARDS patients had RVsD [19]. A recent meta-analysis showed that the prevalence of RVsD was one out of five patients with a large range (from 2% to 51%), probably due to the echocardiographic parameters used to define RVsD [1]. However, the prevalence of RVsD among COVID-19 patients could be influenced by the severity of the disease and was probably underestimated in critically ill COVID-19 patients.

### 4.2. RVD Dysfunction and COVID-19 Infection

In ARDS, the etiology of RVsD is multifactorial and combines several factors that increase RV afterload. This increase in RV afterload causes uncoupling between the RV and the pulmonary circulation, thus, promoting RV dilatation and RVsD. In CARDS, the increase in RV afterload is probably due to a combination of complex physiopathology factors, some of which are specific to the COVID-19 infection. One pathophysiological hypothesis is that COVID-19 infection leads to pulmonary vascular damage, a so-called “acute vascular distress syndrome” (AVDS) [20]. In AVDS, the increase in pulmonary blood flow is favored by pulmonary vessel dilatation and pulmonary neoangiogenesis, leading to perfusion abnormalities toward areas of healthy and diseased lungs [21], resulting in a worsening ventilation-perfusion mismatch and clinical hypoxemia [22]. The increase in pulmonary blood flow increases the RV diastolic overload, inducing RV dilatation and dysfunction. Caravita and al. compared patients with CARDS and with ARDS from other causes and found that cardiac output increased, whilst pulmonary vascular resistance decreased in CARDS patients. These findings support the AVDS hypothesis and RV adaptation in COVID-19 patients [23]. Our study did not find any association between RVsD and myocardial biomarkers, even in the ACP subgroup. Cardiac injury due to COVID-19 infection by direct viral myocardial involvement or myocardial infiltration has been supported by elevated levels of myocardial biomarkers (troponins or natriuretic peptides) [24]. However, Van den Heuvel et al. showed that RVsD was not associated with troponin or NT-proBNP in a cohort of COVID-19 patients. [25]. Our result is probably due to the timing of sampling, as the collection of biomarkers was performed only on the day of echocardiographic examination (within 48 h of ICU admission). Late complications with myocardial injury may have arisen during a prolonged ICU stay.

### 4.3. RV-LSF and Mortality 

The most relevant finding of our report was that RVsD defined by an RV-LSF of <20% was associated with 30-day mortality in CARDS patients. RVsD is a well-known deadly complication of non-CARDS. Several studies observed that CARDS patients with RVsD had higher mortality than patients without RVsD. In a recent meta-analysis, RVsD was associated with a significantly increased likelihood of all-cause of death (odds ratio 3.32 95% CI (1.94–5.70)), but depended on which parameter was used to assess RV systolic function [1]. 

An RV-LSF of <20% was confirmed as an independent mortality factor in our Cox model, contrary to RV-FAC and RV-FWLS. This result may be explained by the fact that RV-LSF is more accurate for identifying RVsD [8] than RV-FWLS, as previously shown [10]. At the beginning of the COVID-19 pandemic, Lie et al.showed that RV-FWLS, with a cut-off value of −23%, was a powerful predictor of higher mortality in COVID-19 disease [7]. Despite a promising future, impaired RV-FWLS seemed to be a common finding [6] and was not associated with mortality in several studies [6,26]. Several factors (loading conditions, RV chamber geometry and cardiac desynchronization) affect myocardial contractility, and hence strain values. This may explain these contradictory results for RV-FMLS [18]. Moreover, RV-LSF allows a global measurement of RV systolic function (not only longitudinal but also radial myocardial contraction), whilst longitudinal strain (RV-LWS) explores only one way of myocardial contraction [18]. 

### 4.4. Bi-Ventricular Dysfunction

Left ventricular dysfunction and RVsD are common in COVID-19 patients [27] because the two ventricles share the same septum. In the RVsD group defined by RV-LSF, LVEF was significantly impaired, contrary to the RVsD group defined by RV-FWLS or RV-FAC. Indeed, the ventricular interdependence complicates the diagnosis of RVD by conventional parameters in case of global dysfunction, notably in the presence of an acute cor pulmonale. RV-LSF can identify patients with left ventricular systolic dysfunction by assessing the following factors: (via the septal point) the contraction of the interventricular septum and (via the apex point) the contraction of the LV apex because of LV and RV apex tethered myocardial fibers [28]. Moreover, RV-LSF was more accurate than RV-FWLS for assessing RVD in CARDS patients with ACP [10]. Immunothrombosis significantly contributes to the pathophysiology, severity, and mortality of COVID-19 disease [29]. There was no difference in pulmonary embolism diagnosis between the two groups during the TEE exam. However, pulmonary embolism during the follow-up was higher in the RVD group. RVD is associated with a significant risk of thrombotic complications, due to thrombi formation in the cardiac chamber and blood vessels [30]. RVD increased the risk of venous blood stasis leading to thrombosis and pulmonary embolism [30]. 

## 5. Limits

We acknowledge several limitations in our study. First, the limited sample size and the monocentric design may have led to underpowered statistical analyses. However, our cohort size is similar to that of other studies published on the same topic [6,31,32]. Concerning the monocentric design of the study, the heterogeneity of the software and measurement techniques for speckle-tracking parameters make it difficult to carry out multicentric studies in critical care. Recent guidelines call for the standardization of 2D-STE measurements to overcome this problem [33]. Secondly, we preferred TEE rather than TTE TEE ultrasound, which is more invasive but is more accurate for RV systolic assessment or ACP diagnosis because of its better image quality in mechanically ventilated patients. In their study, Bleakley et al. were able to evaluate RV-FWLS by TTE in only 56% (*n* = 51/90) of critically ill patients with COVID-19 [31], while only one patient was excluded for insufficient image quality to measure the 2D-STE parameters in our study. Thirdly, in ARDS studies, RVsD echocardiographic definition was based on the following various parameters: many authors used TAPSE or RV-FAC, whilst others used the RVEDA/LFEDA ratio [15,19,31]. We chose RV-LSF to define RVsD for several reasons, which are as follows: RV-LSF is simple, highly feasible and has excellent reproducibility [10]. Moreover, it is considered as the most accurate 2D parameter for RV systolic function assessment [10]. Finally, we did not analyze several parameters that may have influenced 30-day mortality, such as COVID-19 variants, vaccines, and specific COVID-19 therapy during the inclusion period. Further studies are needed to investigate these associations. 

## 6. Conclusions

In moderate-to-severe CARDS, RVsD defined by an RV-LSF of <20% in TEE occurs in 43% of the patients. Contrary to RV-FWLS and RV-FAC, an RV-LSF value of <20% seems to be associated with 30-day mortality. Early diagnosis of RVsD by the RV-LSF could allow therapeutic optimization to improve the prognosis of CARDS. Further studies in ICU focusing on this promising 2D-STE parameter are required.

## Figures and Tables

**Figure 1 jcm-11-02625-f001:**
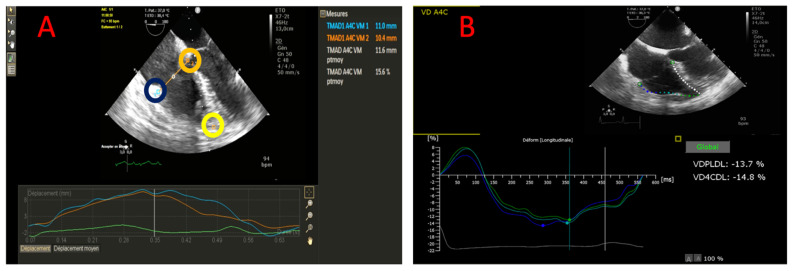
Measurement of 2D-STE parameters in a mid-esophageal four-chamber view. (**A**) TAD. A lateral point (blue circle) and a septal point (orange circle) were placed at the bottom of the RV free wall and the bottom of the interventricular septum. A third point was placed at the apex (yellow circle). TAD lateral, TAD septal and RV-LSF (%) value were automatically displayed. The mid-annular point is automated selected by the software. (**B**) region of interest was generated automatically and adjusted manually. RV-FWLS and RV-GLS were calculated automatically by the software.

**Figure 2 jcm-11-02625-f002:**
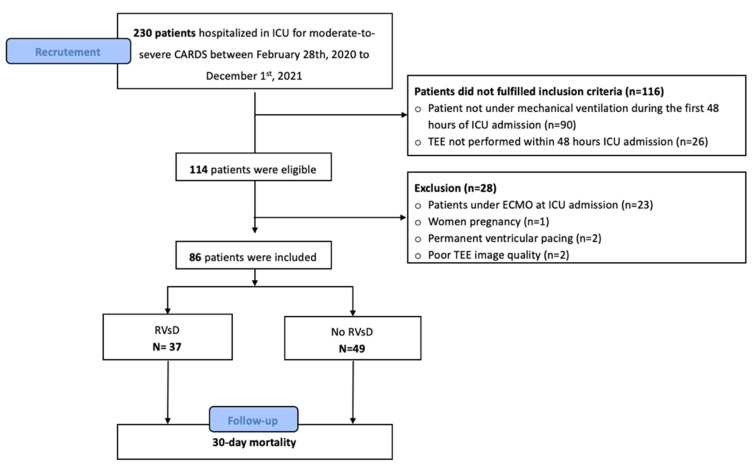
Flow chart of study population. CARDS: acute respiratory distress syndrome related to COVID-19 infection; ECMO: extracorporeal membrane oxygenation; ICU: intensive care unit; RVsD: right ventricular systolic dysfunction; TEE: transesophageal echocardiography.

**Figure 3 jcm-11-02625-f003:**
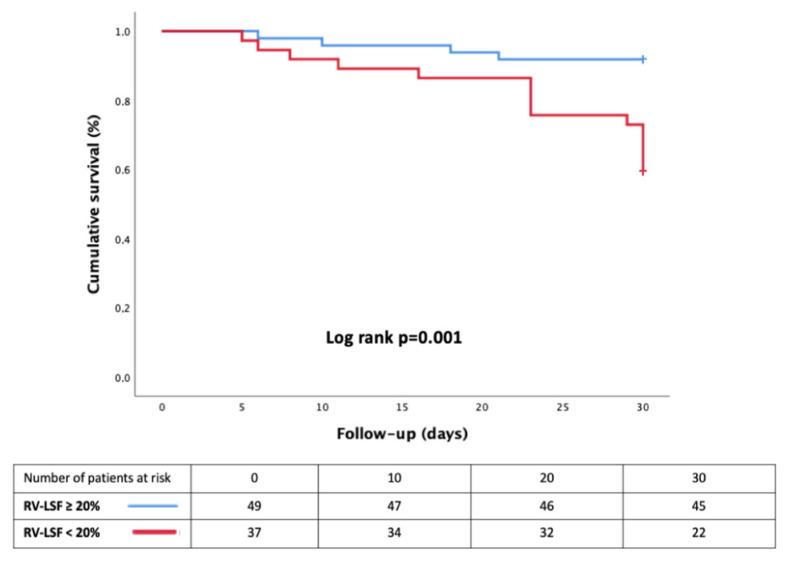
Kaplan–Meier survival curves, according to the presence of RVsD defined by an RV-LSF of <20%.

**Table 1 jcm-11-02625-t001:** Demographic, biological and computed tomography data before TEE.

Variables	No RVsD (*n* = 49)	RVsD (*n* = 37)	*p*
Age (years)	63 (59–69)	59 (55–68)	0.13
BMI (kg·m^−2^)	29.3 (25.8–34.4)	30.1 (24.8–35.8)	0.87
Male gender (*n*; %)	35 (71)	27 (73)	0.47
SAPS II score	45 (29–66)	51 (30–63)	0.94
Medical history, *n* (%)			
No history	7 (14)	5 (13)	1
Hypertension	30 (61)	16 (43)	0.12
Diabetes	15 (30)	8 (21)	0.46
Dyslipidemia	12 (24)	15 (41)	0.16
Smoking (former or active)	6 (12)	6 (16)	0.75
Chronic kidney disease	4 (8)	4 (11)	0.72
COPD/asthma	4 (8)	7 (19)	0.19
Coronary or peripheral artery disease	5 (10)	4 (11)	1
CT scan (*n* = 86/86), *n* (%)			
Ground-glass opacification	42 (85)	36 (97)	1
Consolidation	25 (51)	22 (59)	0.81
Crazy paving	15 (31)	7 (19)	0.21
Lung involvement > 50%	22 (44)	19 (51)	0.66
Pulmonary embolism	1(3)	4 (10)	0.16
Biological data before TEE			
Lactate (mmol^−1^)	2.0 (1.7–2.4)	2.1 (1.5–2.5)	0.56
Serum-creatinine (µmol·L^−1^)	69 (58–88)	86 (69–107)	0.07
BNP (pg·mL^−1^)	53 (32–110)	59 (18–209)	0.79
Troponine Tc HS (ng·mL^−1^)	24 (11–51)	34 (10–66)	0.27
Procalcitonin (µg·L^−1^)	0.54 (0.19–1.72)	0.55 (0.22–2.26)	0.93
C reactive protein, mg L^−1^	181 (96–263)	156 (90–220)	0.72
Time from first symptoms to ICU admission (days)	8 (6–11)	7 (4–9)	0.60

Data are presented as median (interquartile range) and number (percentage). BMI: body mass index; BNP: brain natriuretic peptide; CT: computerized tomography; COPD: chronic obstructive pulmonary disease; SAPS: simplified acute physiology score; TEE: transesophageal echocardiography.

**Table 2 jcm-11-02625-t002:** Hemodynamic, ventilatory and echocardiographic data.

	No RV Dysfunction (*n* = 49)	RV Dysfunction (*n* = 37)	*p*
Hemodynamic parameters during TEE			
Heart rate (bpm)	82 [72–92]	82 [71–97]	0.89
Systolic blood pressure (mmHg)	131 [112–151]	124 [109–141]	0.21
Mean blood pressure (mmHg)	85 [71–96]	84 [70–98]	0.78
Diastolic blood ressure (mmHg)	66 [55–78]	78 [60–80]	0.28
Ventilator settings during TEE			
Tidal volume (mL·kg^−1^)	5.9 (5.5–6.8)	6.0 (5.3–6.6)	0.75
PaO_2_/FiO_2_ (mmHg)	103 (80–167)	110 (90–168)	0.59
Positive end expiratory pressure, (cmH_2_O)	12 (10–14)	12 (10–14)	0.62
Respiratory rate	27 (24–31)	28 (24–30)	0.81
Plateau pressure (cmH_2_O)	26 (23–28)	27 (24–30)	0.23
Driving pressure	14 (11–16)	14 (12–17)	0.37
Respiratory system Compliance (mL·cmH_2_O^−1^)	30.3 (28.1–36.2)	33.8 (29.3–38.3)	0.26
Rescue therapy			
Neuromuscular blocker, *n* (%)	49 (100)	37 (100)	1
Inhaled nitric oxide, *n* (%)	31 (63)	22 (59)	0.53
Vasopressor use, *n* (%)	24 (49)	23 (63)	0.28
- Norepinephrine, (µ/kg/min)	0 (0–0.16)	0.15 (0–0.61)	0.01
Vasoactive-inotropic score (VIS)	0 (0–16)	15 (0–61)	0.01
SOFA cv	0 (0–4)	4 (0–4)	0.02
TEE parameters			
RV EDA	18.5 (14.9–22.0)	22.4 (18.7–26.7)	0.006
RV ESA	9.7 (7.5–12.0)	13.0 (11.2–18.4)	0.001
RV EDA/LV EDA	0.68 (0.56–0.88)	1.06 (0.71–1.15)	0.003
RV-FAC (%)	48.7 (41.1–54.7)	41.2 (32.0–46.9)	0.003
Acute cor pulmonale	10 (20)	18 (49)	0.005
- BNP (pg·mL^−1^)	19 (11–52)	39 (16–105)	0.21
- Troponine Tc HS (ng·mL^−1^)	41 (21–83)	59 (25–115)	0.57
- Lactate (mmol^−1^)	1.8 (1.0–2.1)	1.9 (1.1–2.4)	0.46
Left ventricular ejection fraction (%)	65.6 (57.3–72.0)	56.1 (42.9–67.9)	0.03
Cardiac output (L·min^−1^)	5.0 (4.5–6.5)	4.4 (2.9–6.7)	0.06
Valvular heart disease			
- Severe mitral regurgitation	1	1	-
- Severe aortic regurgitation	1	0	-
2D-STE parameters (*n* = 81/86)			
RV-GLS (%)	20.7 (16.9–27.5)	17.9 (13.2–20.7)	0.005
RV-FWLS (%)	25 (20.5–29.8)	20.2 (16.4–25.6)	0.002
TAD parameters			
❖ TADlat (mm)	23.0 (20.5–26.7)	15.7 (12.0–18.1)	0.0001
❖ TADsep (mm)	14.0 (10.0–15.6)	8.1 (7.2–10.3)	0.0001
❖ RV-LSF (%)	26.0 (23.0–29.4)	16.5 (13.7–19.4)	0.0001

Data are presented as median (interquartile range) and number (percentage). CV: cardiovascular; ECMO: extracorporeal membrane oxygenation; EDA: end-diastolic area; ESA: end-systolic area; ICU: intensive care unit; LV: left ventricle; SOFA: sepsis organ failure assessment; RV: right ventricle; RV-FAC: right ventricle fraction area change; RV-GLS: right ventricle global longitudinal strain; RV-FWLS: right ventricle free wall longitudinal strain; RV-LSF: right ventricle longitudinal shortening fraction; TAD: tricuspid annular displacement; TEE: transeosophageal echocardiography.

**Table 3 jcm-11-02625-t003:** Clinical outcome during ICU stay.

	No RV Dysfunction (*n* = 49)	RV Dysfunction (*n* = 37)	*p*
Outcomes †			
Ventilator acquired pneumonia	38 (76)	32 (65)	0.56
Renal replacement therapy	13 (26)	12 (31)	0.63
Pulmonary embolism	3 (6)	8 (22)	0.04
Cardiogenic shock	2 (4)	7 (19)	0.03
Veno-venous ECMO *	7 (14)	8 (22)	0.41
Veno-arterial ECMO *	0	2 (5)	0.18
Time under mechanical ventilation	17 (11–28)	20 (11–31)	0.70
30-day mortality (*n*, %)	4 (8)	15 (40)	0.0001
Length of stay in ICU (days)	21 (15–44)	23 (11–35)	0.42
In-hospital mortality (*n*, %)	12 (24)	17 (46)	0.04
Hospital length of stay (days)	32 19–49	39 24–55	0.35
In-hospital mortality causes (*n*, %)			
Cardiogenic shock	2	5	0.13
Respiratory failure	3	5	0.28
Multiple organ failure	5	6	0.51
End of life decision	2	1	1

Data are presented as median (interquartile range) and number (percentage). † Data regarding the totality of ICU stay. * Patients requiring ECMO after enrollment due to respiratory or hemodynamic failure. Veno-venous ECMO was implanted according to EuroELSO criteria. ECMO: extracorporeal membrane oxygenation; ICU: intensive care unit.

**Table 4 jcm-11-02625-t004:** Univariate and multivariate Cox analysis of variables associated with 30-day mortality.

Variables	30 Days Mortality			
	Univariate Analysis		Multivariate Analysis	
	HR (95% CI)	*p*	HR (95% CI)	*p*
SAPS II (for each point)	2.9 (1.1–7.8)	0.03	1.03 (1.01–1.04)	0.04
Acute cor pulmonale	3.44 (1.33–8.98)	0.01	3.01 (1.13–7.94)	0.03
PaO_2_/FiO_2_ < 150 mmHg	1.9 (0.55–6.59)	0.29	-	-
Driving pressure > 18	2.14 (0.68–6.68)	0.19	-	-
RVsD				
∘ RV-LSF < 20%	5.51 (1.82–16.7)	0.002	4.45 (1.43–13.8)	0.01
∘ RV-FAC < 35%	0.72 (0.21–2.5)	0.61	-	-
∘ RV-FWLS < 21%	1.38 (0.56–3.4)	0.48	-	-
Pulmonary embolism before TEE	0.81 (0.11–6.11)	0.84	-	-
SOFA cv	1.29 (0.99–1.68)	0.06	-	-

CI: confidence interval; CV: cardiovascular; HR: hazard ratio; SAPS: simplified acute physiology score; SOFA: sepsis organ failure assessment; RV: right ventricle; RV-FAC: right ventricle fraction area change; RV-FWLS: right ventricle free wall longitudinal strain; RV-LSF: right ventricle longitudinal shortening fraction; TEE: transeosophageal echocardiography.

## Data Availability

The datasets used and/or analyzed during the current study are available from the corresponding author upon reasonable request.

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
