# Peer review of "Association between the Right Ventricular Longitudinal Shortening Fraction and Mortality in Acute Respiratory Distress Syndrome Related to COVID-19 Infection: A Prospective Study"

_jcm, 2022, doi:10.3390/jcm11092625_

Round 1

Reviewer 1 Report

Beyls et al. investigated echocardiographic parameters for right ventricular systolic dysfunction and failure. The found that right ventricular longitudinal shortening fraction independently predicts mortality in their cohort. The authors have to be commended for this labor-intensive study.

However, after reading the manuscript I would like to make several comments:

  1. Why were patients on ECMO excluded? ECMO adresses major mechanisms for the development of ACP (hypercapnia, hypoxia, high airway pressure due to mechanical ventilation). Do the authors have data on ECMO patients?

  1. Please provide the vasoactive inotropic score at the time of measurement in addition to the SOFA cv. Please provide the dose of the individual catecholamines at the time of measurement. Can you provide hemodynamical data like blood pressure, heart rate, cardiac output, systemic vascular resistance, central venous pressure in table 2 so the reader can understand the clinical situation during the measurement.

  1. How many patients were graded with RVsD due to RV-FAC < 35%, RV-FWLS < 20% and RV-LSF < 20%. The figure in the supplement should be improved. Do they differ in their clinical characteristics according to table 1 and 2?

  1. on the definition of ACP: How many of the patients categorized as ACP had venous congestion? 

  1. can you comment on the causes of mortality in table 2?

  1. can you provide a post hoc power analysis of RV-LSF so the reader can appreciate the impact of your results?

  1. it is surprising that troponin, natriuretic peptides and lactate are not associated with RVsD. could you provide troponin, natriuretic peptide and lactate for patients with ACP?

Minor: 

line 278: please correct the sentence

line 287 please provide the missing reference

line 290 please correct the sentence

Reviewer 2 Report

The topic is interesting and the work is well described. However, some comments are present:

  • what mortality was investigated (cardiac or all-cause?)
  • how was the follow-up carried out? (ambulatory visits/ telephone?)
  • In the first sentence of materials and methods (Population paragraph) please add that the study included COVID-19 patients. 
  • It may be of interest for the readers to know the total length of stay, including in-ward stay after ICU. Moreover, the authors should add the in-hospital mortality. 
  • Please check the abbreviations. What does ACP mean?
  • In the statistical analysis paragrapher,  add that the Kaplan-Meier survival analysis was performed.
  • Tabel 2. Was the ECMO used after the patients were enrolled? Please make it more clear. Authors might use an asterisk to specify this. 
  • Have valvular heart diseases (VHDs) been ruled out? This is an important question that, if present, could significantly affect the prognosis of these patients. In my opinion, data on (at least moderate) VHDs should be reported.
  • The authors found no differences in clinical, laboratory, and CT results between the two groups. How do you explain that RV dysfunction affects acute clinical status if it alters outcomes at 30 days? The authors indicated that an early time of sampling for biomarkers may have influenced the results. However, this does not justify the results from a pathophysiological point of view unless there are further late complications that may have arisen during hospitalization. Please comment these concerns.  
  • Check reference citation (page 9 line 287) and list (e.g. 11)
  • Conclusions are a summary of the results. Please make them more fascinating by adding clinical implications of your findings.

Round 2

Reviewer 1 Report

I thank the authors for their kindness and have no further remarks.